

# LUCI-EntEx v1.0: A GIS-based algorithm to determine stream entry and exit points at boundaries of any given shape.

Bethanna Jackson[1], Rubianca Benavidez[1], Keith Miller[1,2], Deborah Maxwell[1,3]

[1] School of Geography, Environment and Earth Sciences, Victoria University of Wellington, Wellington 6022, New Zealand
[2] now at Kāpiti Coast District Council, Private Bag 60601, Paraparaumu 5254, New Zealand
[3] now at WSP, Majestic Centre, 100 Willis Street, Wellington, New Zealand

*Correspondence to*: Bethanna Jackson (Bethanna.Jackson@vuw.ac.nz)

**Abstract.** Increasing attention is turning to moderating the impact of human activity on the environment, both to preserve the intrinsic value of ecosystems and species for their own sake, and to protect the benefits we derive from nature for future
generations. Internationally, various regulations and policies are in place or in development to improve our stewardship of the environment and develop more sustainable and resilient management practices. However, policies formulated at national or regional scales are not always suited to enacting targeted and cost-effective approaches at the local scale due to geoclimatic, topographical, or management constraints. The direct monitoring of the local and upstream impacts of every management unit to determine their "net impact" is a costly practice, thus emphasising the need for modelling approaches to complement limited
on-ground measurements. This paper describes and demonstrates tools (*LUCI-EntEx v1.0*) that automatically identify the fluvial and terrestrial flow of water in and out of a study area, such as a river that enters a farm that is impacted by upstream management, or terrestrial flow coming from neighbouring property. By identifying the stream entry/exit points, the "net impact" of land management within the study area can be more easily quantified based on the contribution of neighbouring and upstream areas, aiding in the decision-making process. This algorithm also facilitates the identification of inconsistencies
in data such as differences between the legal/official catchment boundaries and the hydrological boundaries determined by the representation of terrain and river networks. If such inconsistencies are not resolved, they can cause further error propagation in later stages of the modelling process. Four case studies of New Zealand management units - two at the farm scale and two at the catchment scale - demonstrate the algorithm's utility in determining fluvial and terrestrial entry/exit points and highlighting potential data inconsistencies. The farm case studies also use the Land Utilisation and Capability Indicator (LUCI)
framework to demonstrate how this algorithm can be embedded in other models for further value: in this case, we show its potential to improve predictions and enhance management of nutrients and sediment.

## 1 Introduction

Over the last two decades, increasing attention has been paid to understanding how targeted land management might mitigate
flood risk, improve water supply, water quality, erosion and sediment outcomes, and aid other environmental benefits. In



New Zealand and elsewhere, farmers, foresters and other land managers are facing both policy incentives (carrots) and regulatory demands (sticks) designed with the intention of increasing the aforementioned environmental benefits and reducing risk from environmental hazards (Quinn et al., 2009; McDowell et al., 2016). However, policy incentives and regulations formulated at national or regional scales are often very broad brush and not always suited to the local

environment's geoclimatic, topographic, and management constraints (Schröder et al., 2004). To better understand the impact of local land management on environmental as well as socio-economic outcomes, and inform more cost-effective and targeted policy, we need to be able to distinguish between local and upstream impacts to observe the "net impact" on the local landscape (McDowell et al., 2016). The cost of directly monitoring this at every farm or forestry unit is prohibitive; therefore, modelling approaches are needed to complement the necessarily limited on-ground measurements.

Creating a hydrologically and topographically consistent digital elevation model (DEM) with an appropriate stream network is an important part of modelling landscapes for hydrological applications. The identification of the stream network is particularly important in understanding transport of water, sediment, nutrients and other mass through a landscape. GIS toolboxes have algorithms in place to help delineate catchments and subcatchments in order to model the movement of water

through a landscape (Maidment, 2002). However, these tools are generally used to understand flow pathways within an isolated catchment or subcatchment, where there is no water transfer from outside the catchment boundary and there is often only one significant outlet to consider. Complexities arise when the area to be modelled covers only part of a catchment or encompasses several subcatchments with multiple entry and exit points along its boundary. This is the case with many farms, forestry units, and other land management units that have been defined according to administrative boundaries rather than

natural catchment and subcatchment boundaries.

In the context of partial catchment studies within a larger watershed or that span several watersheds, there may be several locations where the stream network is entering or exiting the area being modelled. In rural areas, for example, farm boundaries do not always align with watershed extents or the stream network. As such, a farm may span several water courses resulting

in the stream network crossing the farm boundary in numerous locations. Further, a farm along the coastline may be wide enough to encompass the lower reaches of more than one river catchment, but its boundary may not extend into the headwaters of these catchments. In both cases, the stream network may intersect the farm boundary at several locations. It is also not uncommon for streams, rivers, or other water bodies to form part or all of the boundary between "owned" parcels of lands, farms or otherwise. Depending on the accuracy of the farm boundary or other area of interest being modelled, and the

consistency of the input stream and topographical information with the actual stream and topography, a river may enter and exit the modelling domain boundary at several locations.

While it may be relatively easy to identify by human eye the direction of flow and entry/exit points as a river moves through a study area or along its boundary, it is more difficult to determine computationally. In this paper, we describe a GIS-based

process for automating the identification of fluvial entry/exit points on the modelled boundary. In being able to identify these points, the impact of activities within this boundary on neighbouring and downstream areas as well as the contribution of nutrients, sediment and other ecosystem services from adjacent areas can be more easily quantified and subsequent land management decisions more robustly determined. This process also facilitates the identification of inconsistencies in input

data such as differences in legal/official catchment boundaries relative to those implicit in the digital representation of terrain and river networks. These inconsistencies, unless resolved, can cause further error propagation in the later stages of the modelling process, and diminish the utility of models to inform scientifically robust and appropriate policy. In *Sect. 3*, case studies at both farm and catchment scale demonstrate the utility of this algorithm to estimate "net impacts" of on-farm management and to identify input data inconsistencies.

## 10  2 Solution/methodology

A procedure for identifying landscape entry/exit points has been developed and applied using code embedded in the Land Utilisation and Capability Indicator (LUCI) framework which is further explained in *Sect. 3*. This paper discusses the algorithms for preprocessing the input DEM and stream network, identifying fluvial entry/exit points, and identifying terrestrial flow using a standalone version of the code (*LUCI-EntEx v1.0*). Within *LUCI-EntEx v1.0*, the user is only required to input

the DEM, stream network, and a mask of either the study area in isolation or the study area including uphill and/or upstream contributing areas. These inputs are enough to produce the required flow direction (*fdr*) and flow accumulation (*fac*) rasters required for the entry/exits algorithms. The toolbox requires ArcGIS 10.4.1 or higher with Spatial Analyst.

**Preprocessing the DEM**

The first tool to run is the *Preprocess DEM* tool. The purpose of preprocessing the DEM is to reconcile inconsistencies between the input DEM raster and the stream network polyline shapefile. Artefacts in the DEM may hinder the identification of hydrological features (e.g. streams), and DEM reconditioning is used to modify DEM cell values in order to better represent the hydrology of the landscape and identify known flow pathways (Callow et al., 2007). Figure 1 shows the steps involved with preprocessing the DEM in order to produce hydrologically consistent outputs of flow direction (*fdr*), flow accumulation

(*fac*), and a stream network. These three files are fundamental inputs to the entry/exits algorithm so the preprocessing step is required in order to address inconsistencies that may lead to errors in subsequent steps.

The inputs for this algorithm are the DEM, a stream network, and a mask. The mask can either be the study area in isolation or the study area including uphill and/or upstream contributing areas. The difference between using a mask of just the study

area and a mask that includes the uphill contributing areas will be shown in the farm-scale case studies in *Sect. 3*. All the input data is first checked and tidied to ensure they all contain information, are in the same coordinate system, the mask is dissolved into one multipart polygon feature, and all inputs are clipped to a buffered version of the mask. The coverage of inputs must





be over a buffered version of the mask as the entry/exit code requires information about the flow accumulation of streams within and outside the study area.

The input stream network is then "burned" into the DEM based on the AGREE method of Hellweger (1997) using the steps
5    shown in Figure 2. "Burning" the streams helps to remove artefacts in the DEM that prevent the accurate modelling of water through the landscape. These artefacts may block water from reaching the streams, thereby not allowing those real-world flow pathways to be extracted from the DEM. The DEM cell values are modified to be more consistent with the input stream network and to direct flow towards the streams (Hellweger, 1997). The burned DEM is further processed to fill spurious sinks, produce the flow direction raster, and the flow accumulation raster.

Information from the flow accumulation raster is then used with the stream initiation thresholds set by the user, which determine where the streams in the output stream network initiate, are located, and how they flow through the landscape. The default initiation thresholds and stream burning parameters provided should be modified depending on the study area's characteristics. If the delineated streams do not accurately depict the real-world hydrological features, these thresholds and
15    parameters can be further calibrated in order to produce a more accurate stream network. The procedure described for extracting the stream network from the DEM is a fundamental step in creating hydrological models and can be done manually in ArcMap using standard Spatial Analyst hydrology tools (Tarboton et al., 1991; Maidment, 2002). The *Preprocess DEM* tool automates this process by chaining these steps together in order to delineate a stream network.



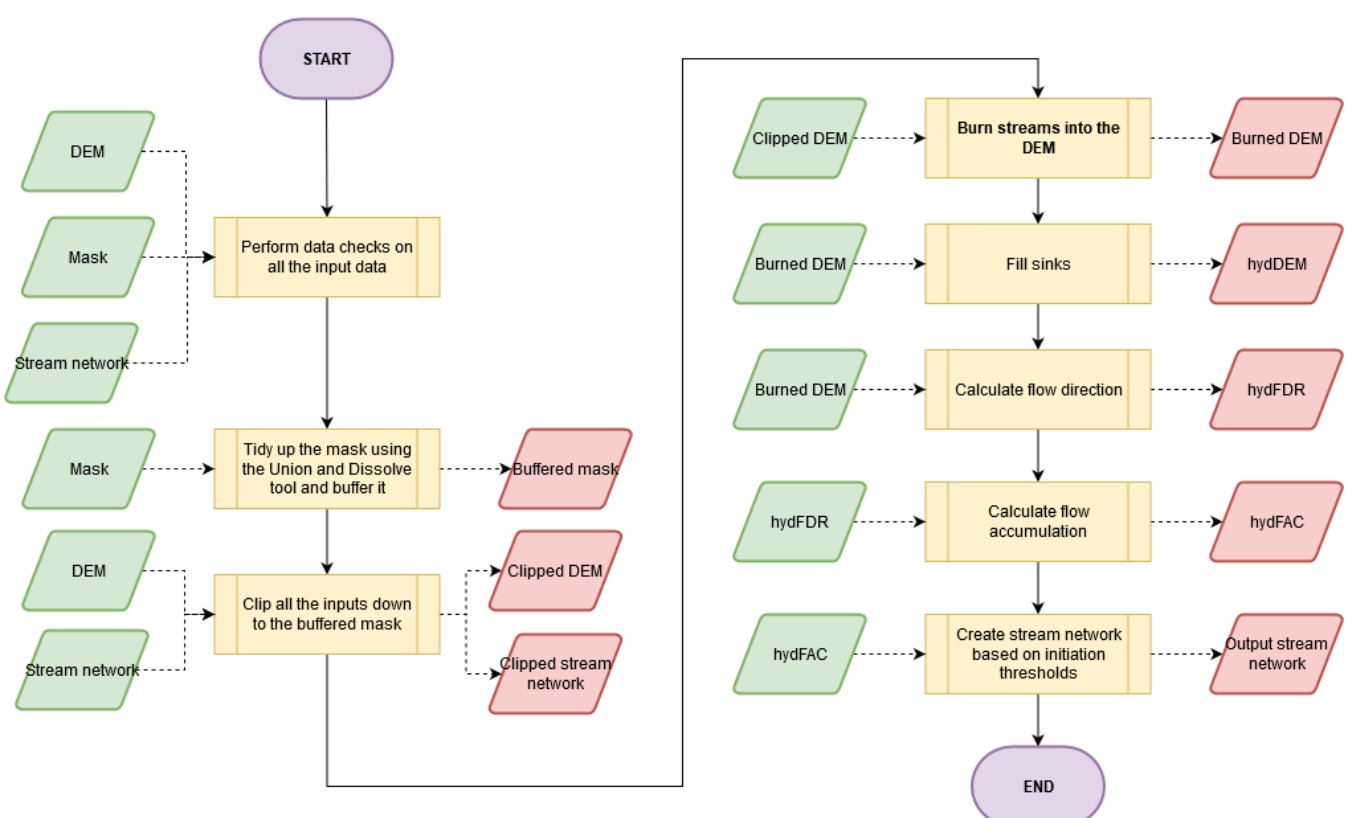

*Figure 1. Flowchart for preprocessing the input DEM and stream network to generate hydrological information and a stream network based on initiation thresholds following the procedures outlined by Maidment (2002). (Purple nodes: terminal ends of algorithm; green parallelograms: inputs; red parallelograms: outputs; yellow boxes: processes.)*

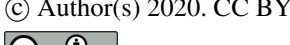



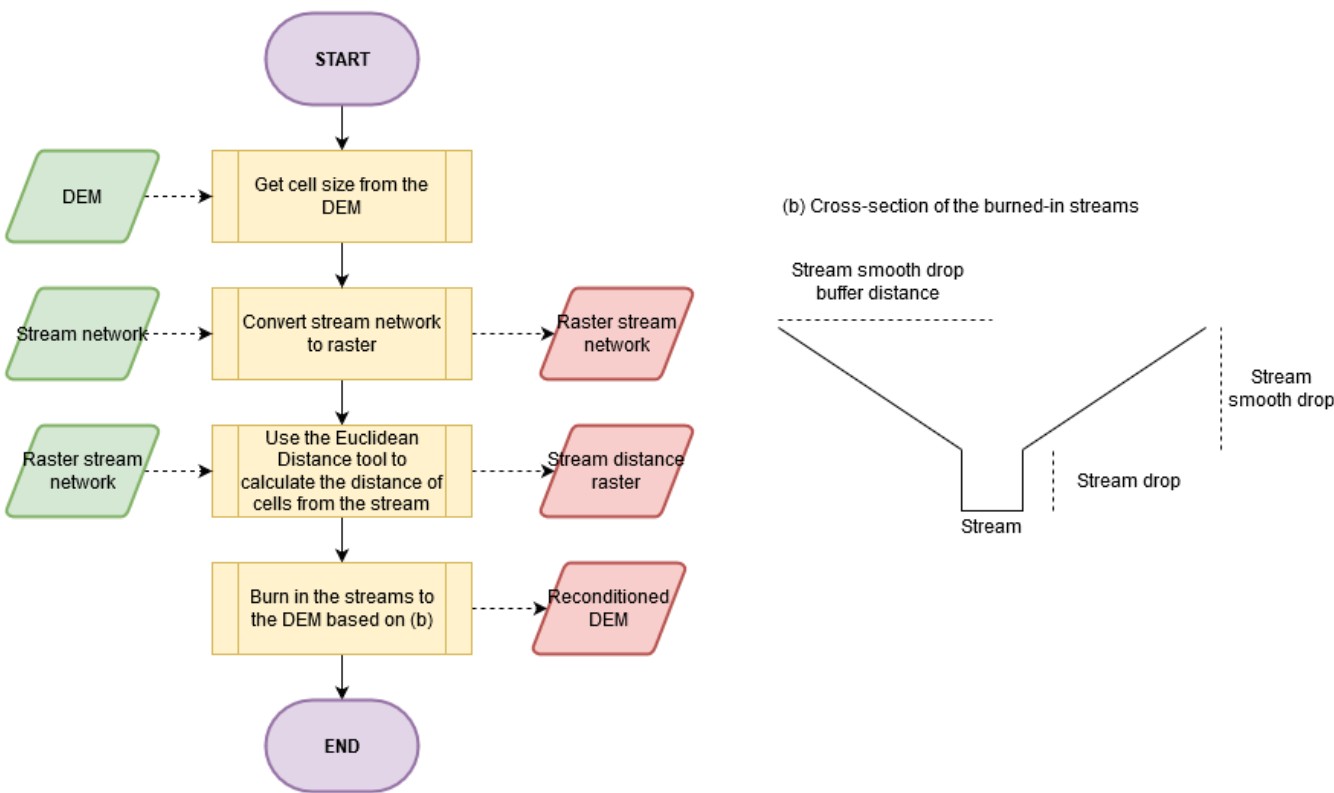

*Figure 2. Flowchart for reconditioning the DEM by "burning" in the streams based on the AGREE method by Hellweger (1997). (Purple nodes: terminal ends of algorithm; green parallelograms: inputs; red parallelograms: outputs; yellow boxes: processes.)*

**Fluvial entry/exit procedure**

After preprocessing the DEM, the resulting stream network and flow accumulation raster (hydfac) are used with the study area boundary to run the Determine stream entry/exit points tool. A breakdown of the steps taken by the entry/exit points tool are shown in Figure 3 and Figure 4. The stream network is separated into reaches (Figure 5a). These curved reaches are then broken into their component straight-line segments; the stream reaches will typically be somewhat curved, but a computer

represents these curves as a series of connected small straight-line segments (Figure 5b). The straight-line segments which intersect the study area boundary are identified. These straight-line segments will contain one or more points where the stream network meets the study area boundary. These intersection points are then classified as entry/exit points using the process described below.

To determine if these intersection points are entry or exit points to the study area, the flow accumulation at the two end points of their associated straight-line segment are examined. If the end point which is located outside the study area has a higher *fac*





than the end point inside the study area then the stream is exiting the area of interest. Conversely, if the *fac* is higher inside the study area then the stream is entering the area of interest. If the *fac* is the same at each of the end points then it cannot be determined if the stream is entering or exiting the boundary. If both ends of the straight-line segment are inside (or both outside) the area, then the point is a vertex with the stream network just touching the boundary and therefore it is treated as if

it is neither exiting nor entering the area of interest (Point 3, Figure 6a).

The above describes the process when there is only one intersecting point on a straight-line segment. While this will be the case most of the time, there may be multiple intersecting points on a single straight-line segment (Point 4b, Figure 6b). In this case, all intersecting points are listed in a table with their associated *fac* and ordered by distance from the 'start' of the line

segment (chosen randomly from the two end points of the straight line). If the 'start' point of the line has a higher *fac* than the 'end' point, and is the located within the study area mask, then the stream segment is entering the area of interest. Otherwise, the segment is exiting the study area mask. The intersection point closest to the 'start' point is then classified as entry/exit, depending on whether the stream is overall entering or exiting the study area. For each subsequent point in the table, the decision as to whether the point is entering or exiting the study area is dependent on whether the previous point is exiting or

entering. If the previous point in the list is exiting, then the following point must be entering the study area.



Determine entry / exit points

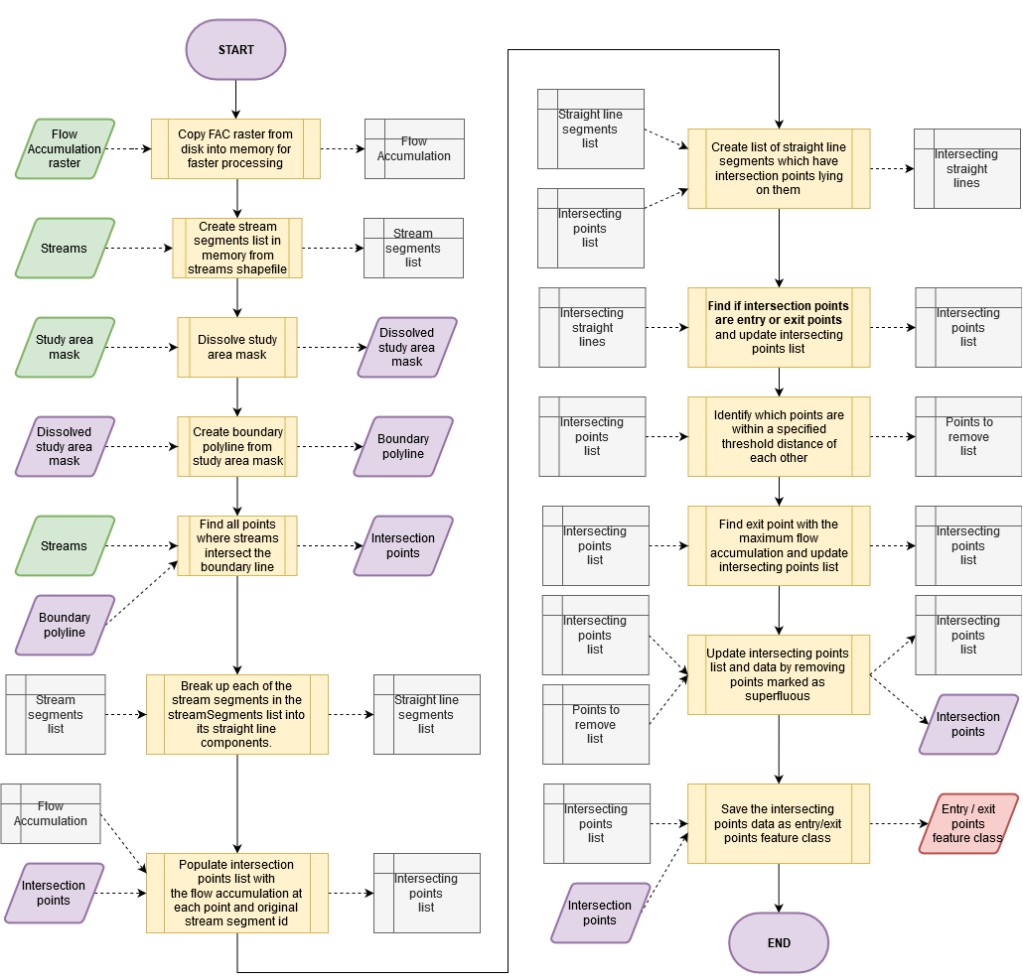

*Figure 3: Flow chart of steps to create the entry/exit points feature class. The procedure for **Find if intersection points are entry or exit points** is further illustrated in more detail in Figure 4.*



*Figure 4: Flow chart showing steps involved in determining in stream segment end points are entering or exiting the area of interest. The numbers in the red circles correspond to the intersection points shown in Figure 6, and in the inset figure.*





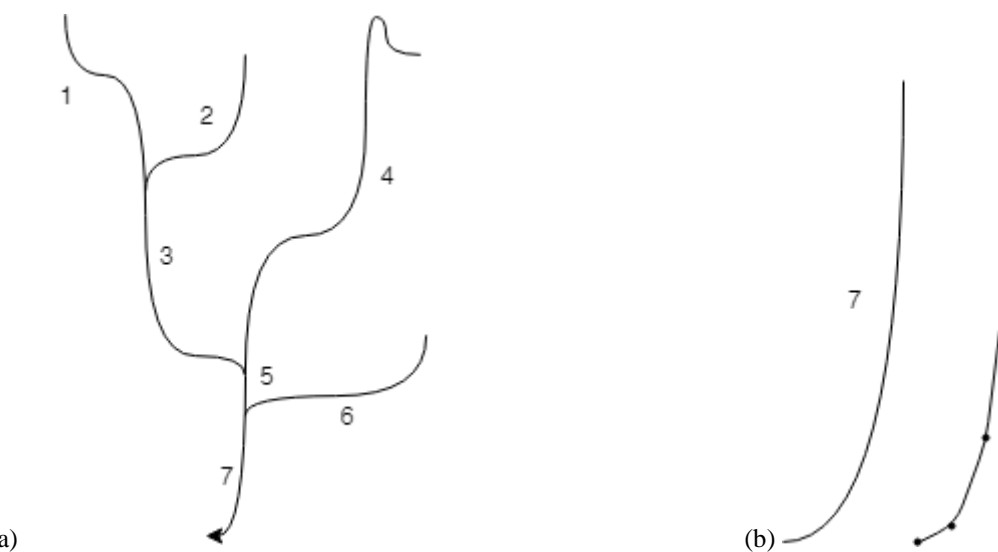

*Figure 5. (a) The stream network is broken up into separate stream segments. (b) The stream network is broken up into straight-line segments and those segments which intersect with the study mask are identified.*

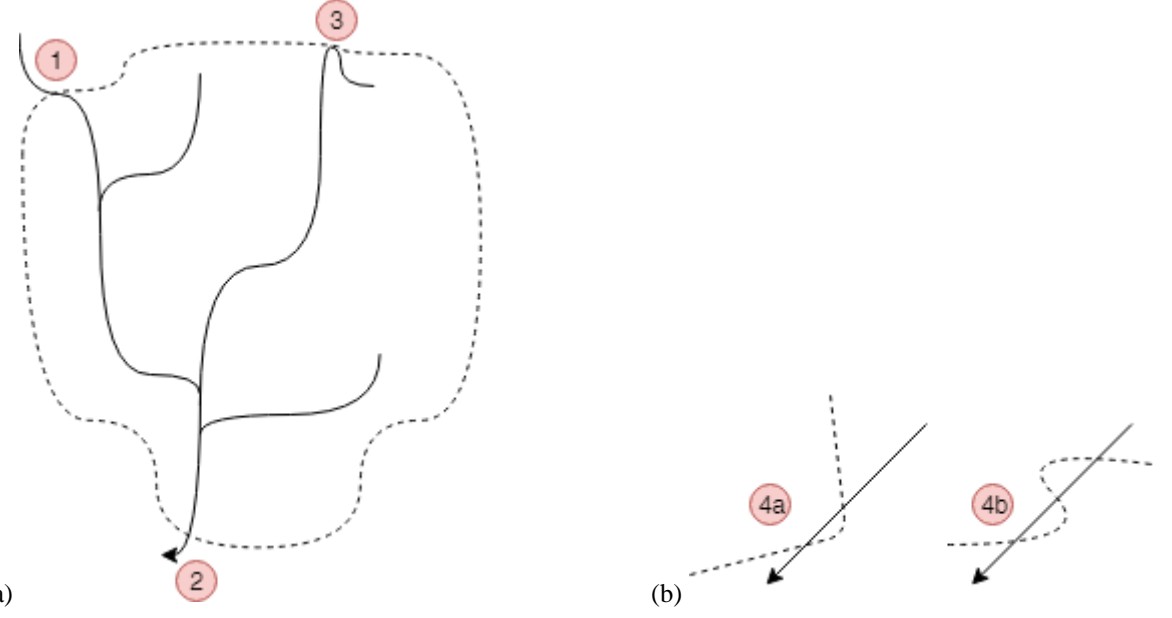

*Figure 6. (a) Different intersection points that intersect with the study area mask. Point 1 is an entry point, while Point 2 is exiting. Point 3 illustrates a vertex where the segments touch the boundary but do not cross it. (b) Straight-line stream segments with multiple intersections.*





Once all intersection points have been classified as entering or exiting the study area, the next step is to remove superfluous entry/exit points that may occur due a stream running along the boundary line of the study area (Figure 7) or due to breaks in the farm boundary caused by infrastructure (e.g. roads). As the code checks each of the entry/exit points, it stores the identifier of any points deemed superfluous into an array that is used later in the code to remove these points. Identifying a point as

5 superfluous is mainly done through checking the coordinates of each of points to find pairs that are less than a specified threshold distance away from each other. In *LUCI-EntEx v1.0*, this threshold distance is set to 100 spatial reference units (e.g. 100m in the case studies) but can be modified in the future or taken in as a user-input parameter depending on study area characteristics. If two points are found within the threshold distance of each other, they are deemed superfluous and added to the array of points to remove.

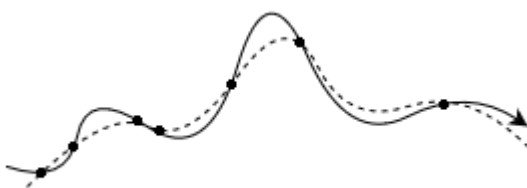

*Figure 7. Example of a stream (solid arrow) running along the boundary of the study area (dashed line) with multiple entry/exit points.*

The penultimate step is to find the primary outlet of the study area. While there may be multiple exit points, the intersection

15 point with the highest flow accumulation is deemed the main exit point in the catchment (Figure 8). It is important to note that only using the study area may incorrectly place the main exit, as the flow accumulations calculated may be underestimated if the streams are part of a larger river or network outside the study area that is beyond the modelling domain. The identifier of this point is recorded and will not be removed in the final part of the code if the code has marked it as superfluous. Finally, the code processes all intersection points, either removing them or keeping them and updating their record with information such

20 as the point number, point type (entry, exit, or main exit), and the stream segment that the point occurs on.



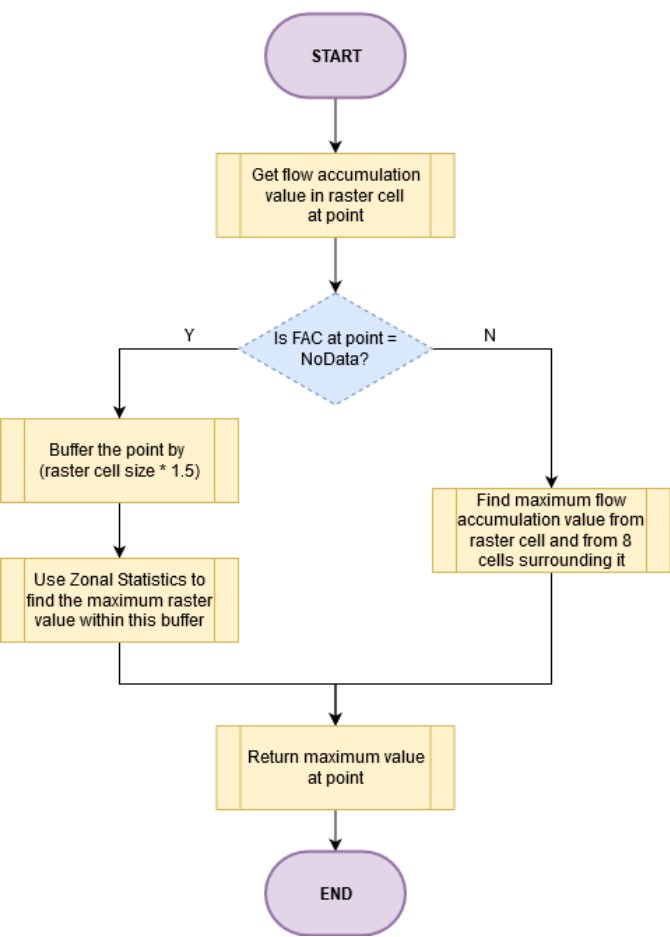

*Figure 8. Steps to identify outlet point of mask.*

**Terrestrial entry/exit procedure**

5    The steps to determine whether a cell's terrestrial flow goes into the study area, out of it, or along a ridgeline are shown in Figure 9. A study area boundary and flow direction raster (*fdr*) are needed. The *fdr* identifies the direction in which flow is moving from cell to cell based on D8 flow direction method (Tarboton, 1997). The algorithm identifies the raster cells that comprise the boundary of the study area and the associated flow direction for each cell. The rows and columns for those boundary cells are stored into an array. For each non-zero element in that array, the flow direction of each element is used to

10    calculate the rows and columns of the cell the water is flowing from (starting cell) and the cell the water would flow into (ending cell). If the starting cell is inside the study area and the ending cell is outside the study area, then water is flowing out





of the study area. The opposite applies for starting cells outside the study area with ending cells inside the study area where water would flow out of the study area. In other cases where the starting and ending cells are both either inside or out of the study area, then the water is flowing along a ridgeline.

## Terrestrial entry / exit points

*Figure 9. Steps to determine terrestrial entry/exit points.*



## 3 Implementation/Application

The procedure and algorithms described in this paper are mainly produced by the *LUCI-EntEx v1.0* code which is freely accessible (see later section on code and data availability). As mentioned before, the entry/exits algorithms are also embedded in the broader codebase of the LUCI framework (*LUCI v0.9*). The LUCI framework is a land management decision support

framework with a variety of modules that often rely heavily on topographical data for subsequent flow-based calculations. LUCI has a proven record of providing reliable output for exploring the impact of land use or management changes on ecosystem service provision in multiple countries, enabling better spatial planning of land management interventions (Emmett et al. 2017; Thomas et al., 2019; Tomscha et al., 2019; Pedersen Zari et al, 2020). Uniquely, LUCI is able to model at a range of spatial scales – from field or plot level up to catchment/watershed or even national level (Sharps et al., 2017; Emmett et al.,

2017, Thomas et al., 2019). It is also capable of comparing different ecosystem services at once, identifying where co-benefits or trade-offs may exist in the landscape.

LUCI requires a DEM (to produce a hydrologically and topographically consistent surface), soils and land cover (for scenario generation) and other inputs (if available) – precipitation, evapotranspiration, stream network. These latter inputs are available

for the study areas and have been used as input across all analyses. LUCI can assess a range of ecosystem services including agricultural production, carbon sequestration, water quantity, water quality, erosion and sediment, amongst others (Tomscha et al, 2019, Sharps et al., 2017, Jackson et al., 2013).

Both versions of the code produce broadly the same stream networks, entry/exit points, and main stream outlets. There are

small differences brought on by the slightly different stream networks they produce. Although both versions of the code preprocess the DEM in a similar manner, *LUCI-EntEx v1.0* does not use a weighted flow accumulation in the stream delineation process. Within *LUCI v0.9*, the code uses additional information about soil, rainfall, and evapotranspiration to produce a water budget and an accompanying rainfall and evaporation-weighted flow accumulation. These differences are further discussed in *Sect. 4*. This paper presents results from two versions of this entry/exits code:

• *LUCI-EntEx v1.0*: the standalone code described in this paper, freely available for use, modification and redistribution;

• *LUCI v0.9*: the code embedded within the bigger LUCI framework which produced the Nitrogen results, highlight the additional value of the entry/exit point identification algorithm when combined with complementary modelling.





The Nitrogen and other chemical routing tools in LUCI use a modified export coefficient modelling approach to estimate non-point source nutrient loads (Trodahl et al., 2017). The export coefficient for each point in the landscape is determined by land cover/use, soil type and slope, climate, and presence of agricultural inputs (e.g. fertiliser, stock, effluent, irrigation). These terrestrial loads of nitrogen are then routed to the stream via various flow pathways (overland flow, throughflow in soil,

groundwater etc.), and intercepted or attenuated where appropriate. LUCI produces output that includes maps showing terrestrial load (kg ha$^{-1}$ yr$^{-1}$), accumulated load (kg yr$^{-1}$) and accumulated concentration (mg L$^{-1}$) to any point in the landscape as well as in-stream nutrient load (kg yr$^{-1}$) and concentration (mg L$^{-1}$). Accumulated values consider not only the load or concentration at that point but also the contributions from uphill or upstream sources within the study area. If the modelling domain also includes the uphill/upstream contributing areas, the accumulated values also account for those contributions. LUCI

also categorises load and concentrations from very low to very high based on user-specified thresholds.

Four case study examples are presented: two farm scale examples and two catchment scale examples. The outputs for the farm scale are used to compare differences in stream network and stream entry/exit points when the upstream contributing area is included rather than a small buffer around the study mask. Farm A covers 225 ha on a relatively flat area in Southland, New

Zealand, while Farm B is located in a headwater catchment in Otago, New Zealand, and covers 448 ha (Figure 10). The main land use for both is pastoral farming. The Mangatarere (15175 ha) and Piako (147910 ha) catchments are both located in the North Island (Figure 10). The terrain of the Mangatarere catchment consists of very steep mountainous terrain in the headwaters with mainly forested areas, and very flat plains with mainly pastoral and agricultural land use. The Piako catchment has complex terrain in the headwaters but becomes very flat as the stream moves down to the Firth of Thames. Like the

Mangatarere, the flat areas of the Piako are mainly pastoral and agricultural land use, but there is a large (>10,000ha) remnant peat dome protected as a stewardship area and wildlife management reserve under the Ramsar convention present in the lower part of the catchment (Ramsar Sites Information Service, 1992).

The 15m NZSoS digital elevation model is used to represent topography for most of the case studies, and a national stream

network obtained from the Ministry for the Environment (NZ) is used as the input stream network (Snelder et al., 2010; Columbus et al., 2011). In the case of the Mangatarere catchment, a finer resolution 5m DEM is used, resampled by the authors from the Wellington LiDAR 1m DEM (LINZ, 2013). For all case studies, the DEM coverage extended beyond the study area boundary to include "uphill" or "upstream" contributing areas for farms, and to allow the code to determine stream entry/exit points for the catchments. Within the main LUCI framework, the DEM was reconditioned to remove any spurious sinks and

to ensure consistency between the stream network and topography (Jackson et al., 2020). Using the standard Spatial Analyst Hydrology tools from ESRI (Greenlee, 1987; Jenson and Domingue. 1988; Tarboton, et al.,1991), flow direction and accumulation fields are calculated and used to test the algorithms. Both *LUCI v0.9* and *LUCI-EntEx v1.0* perform these preprocessing procedures automatically in order.

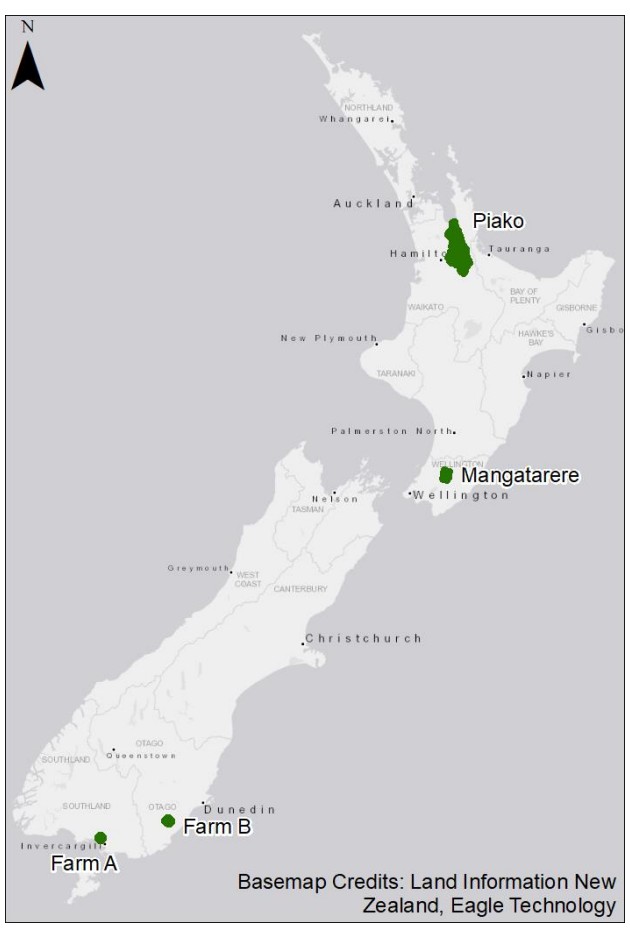

*Figure 10: Location of case study farms and catchments. Farm A is located in Southland, New Zealand. Farm B is situated in Otago, New Zealand. Both are used primarily for pastoral farming. The Mangatarere catchment is located in central Wairarapa. The Piako catchment is located on the Hauraki Plains draining into the Firth of Thames.*

## 4 Results and Discussion

### Fluvial entry/exit procedure

Identifying if a stream is entering or exiting a farm is important to be able to track water, nutrients, sediments and other pollutants through a landscape. In both farm case studies, the study area mask is a partial watershed and has streams entering and exiting at various locations. Applying the methodology above, flow direction and flow accumulation rasters are generated and used to identify if the stream is entering or exiting the farm.

A stream runs through Farm A entering from the northeast and meanders toward the centre of the farm before exiting in the southeast (Figure 11). Provided with just the study area information, it is not immediately obvious as to which direction the



stream is flowing. The outputs are further complicated when the analysis is undertaken using just the study area plus a buffer (Figure 11a) rather than the entire upstream contributing area (Figure 11b). Although there are the same number of entry and exit points intersecting the farm boundary, the main stream appears to be initiated within the farm boundary when just the buffer is used, rather than as a stream that has come in from outside the farm boundary. Because of the limited area being

5 modelled, the generated stream network reflects the fact the accumulation thresholds above which a stream is defined are not met and hence the stream network is less connected or shorter. This has implications for understanding the impact of the farm on the water quality, sediment etc if the upstream contributions are not accounted for. Explicitly accounting for the upstream contributing area results in more physical consistency with an entry point of the stream now identified in the northeast (Point 4, Figure 11). The main exit point remains correctly located in the south-east.

When comparing the results from *LUCI v0.9* (Figure 11) with the results from *LUCI-EntEx v1.0* (Figure 12), the stream networks are broadly the same aside from some minor differences due to LUCI v0.9's direct consideration of rainfall and evaporation . In Figure 11, the exit Point 5 on the western side of the farm area is not present in the results generated using *LUCI-EntEx v1.0* (Figure 12). Both versions of the code identify the correct outlet on the farm (Point 2 in Figure 11 and Figure

15 12).

**Results using LUCI v0.9**

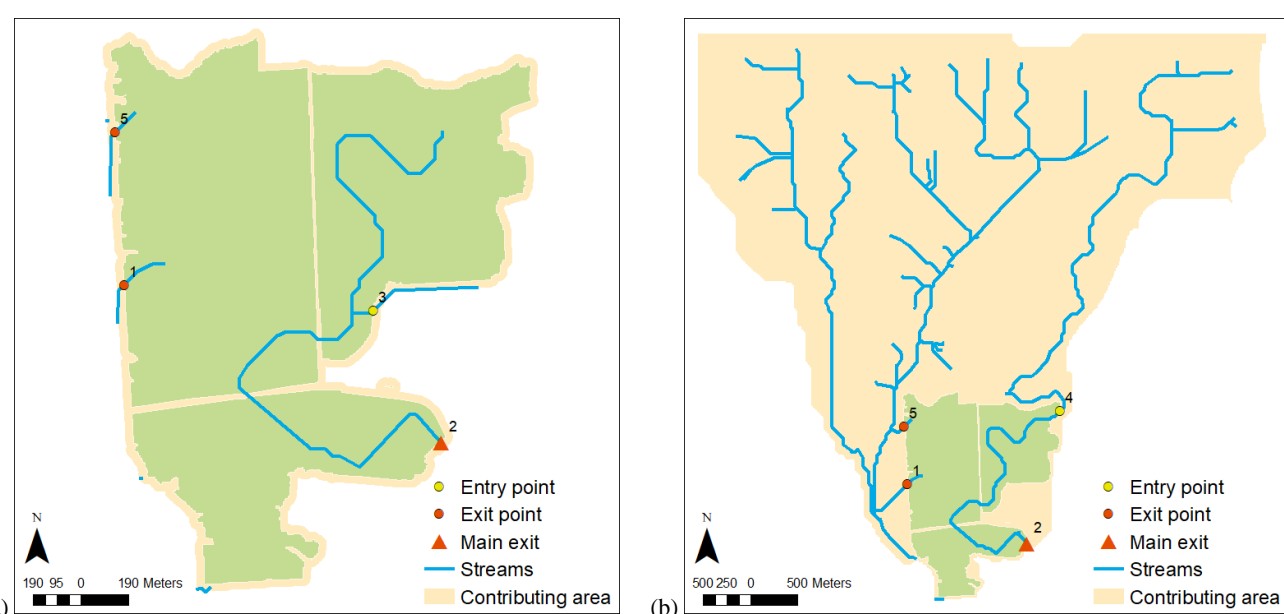

*Figure 11. Farm A (225 ha). Stream entry and exit points generated using the LUCI v0.9 code when using (a) just the study area mask (green) and buffer (beige), and (b) when the upstream contributing areas (beige) are modelled. Note the main stream appears to initiate*



*within the study area boundary. The 'breaks' in the farm boundary indicate a road or other infrastructure which physically separate parts of the farm. The code does not mark these points as entry or exit points since they are still part of the main stream network.*

**Results using LUCI-EntEx v1.0**

*Figure 12. Farm A (225 ha). Stream entry and exit points generated using the LUCI-EntEx v1.0 code when using (a) just the study area mask (green) and buffer (not shown), and (b) when the upstream contributing areas (beige) are modelled.*

There are two main headwater streams that run through Farm B, in various directions. As with Farm A, including the full upstream contributing area provides greater consistency with the actual on-ground stream network (Figure 13b). There are 6 exit points intersecting the farm boundary when only the study area and buffer are considered (Figure 13a), compared with 3 entry points and 6 exit points when the upstream contributing area is modelled (Figure 13b), sensibly reflecting the path of the river as it flows along this boundary. Further, without the upstream areas explicitly accounted for, the main exit is located where the eastern-most stream exits to the southeast (Point 4, Figure 13a). There is one main stream which flows from the centre of the farm toward the west, with another stream which appears to meander in and out of the farm along the northern boundary. Using the study area only, the two streams are disconnected, whereas in reality the two streams converge and continue to meander in and out of the farm toward the actual main exit in the west (Point 7, Figure 13b and Figure 14b). A greater number of entry and exit points are identified due to the connections between the farm and the stream network outside the farm boundary and greater flow accumulations being able to correctly identify the stream network.

Many of the entry and exit points identified in the intermediate steps of the code are within a short distance, where the stream has meandered out and in over a short distance. In many cases, these entry and exit points are spurious and should be ignored.



However, there can be instances where another tributary enters converges with the river just outside the farm boundary before entering back into the study area a short distance downstream. In this situation, the entry and exit points should be retained so that the farm impact can be directly quantified. Our algorithm checks through these entry and exit points, and in our test cases so far, has been able to correctly and automatically identify and remove the spurious points while retaining tributary

convergences. An example is demonstrated in Figure 13b where the northern stream enters at Point 9, exits at Point 1, and entry Point 10 is retained because a new tributary has converged with the river, bringing additional inputs.

As with the results from Farm A, the stream networks delineated by *LUCI v0.9* and *LUCI-EntEx v1.0* for Farm B are broadly the same. Figure 14a shows the results on Farm B using *LUCI-EntEx v1.0* showing more streams delineated on and around the

study area. On the western side of the farm, there are more exit points and one entry point identified due to more stream reaches being delineated on that side. In both cases in Figure 13a and Figure 14a, the main outlet was identified to be on the southeast corner of the farm (Point 4). When the uphill contributing areas are considered (Figure 13b and Figure 14b), the entry/exit points are the same except for one extra exit point (Point 14, Figure 14b). Both versions of the code also identified the same main outlet to the northwest side of the farm (Point 7 in Figure 13b and Figure 14b).

### Results using LUCI v0.9

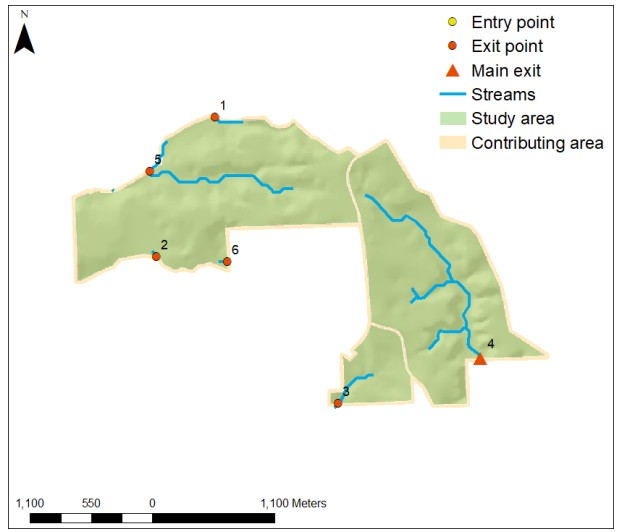
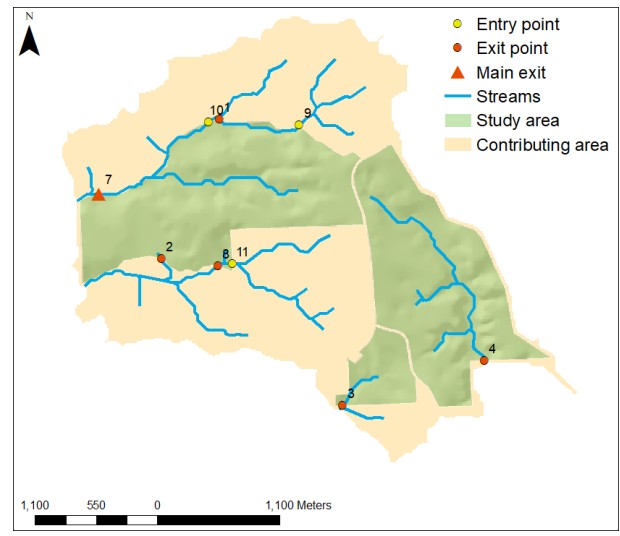

*Figure 13. Farm B (448 ha). Stream entry and exit points generated using the LUCI v0.9 code when using (a) just the study area mask (green) and buffer (beige), and (b) when the upstream contributing areas (beige) are modelled. Note the main stream exit is located in the south-east in (a) while in (b) the two streams in the north are now shown to be connected and therefore the main stream exit point is in the*
*west, reflecting that these two streams together generate a larger streamflow that the stream in the east.*



**Results using LUCI-EntEx v1.0**

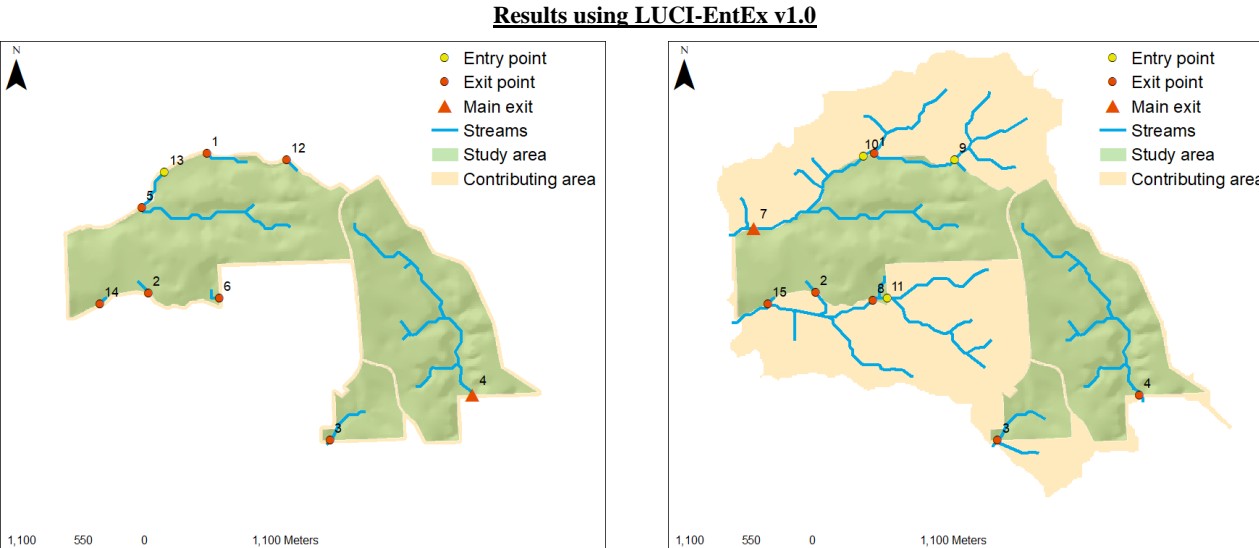

*Figure 14. Farm B (448 ha). Stream entry and exit points generated using the LUCI-EntEx v1.0 code when using (a) just the study area mask (green) and buffer (not shown), and (b) when the upstream contributing areas (beige) are modelled. Note the main stream exit is located in the south-east in (a) while in (b) the two streams in the north are now shown to be connected and therefore the main stream exit point is in the west, reflecting that these two streams together generate a larger streamflow that the stream in the east.*

The outputs also have implications for quantifying the impact of the farm on ecosystem services. In order to assess the impact of land management decisions on ecosystem services it is important that the farm or area of interest can be modelled in isolation from surrounding land uses. Modelling the upstream contributing area helps elucidate the impact of upstream activities, further clarifying the "net impact" the farm has on ecosystem services. For demonstration purposes, we use the stream entry and exit points to quantify the impact the farms have on water quality. LUCI's *Nitrogen* tool calculates nitrogen loads and
concentrations at the different entry and exit points. Note that the results are indicative only; nitrogen loading to the farms (and their upstream areas) is based on regional climate information and national data on land cover and soil only; the numbers do not reflect actual local land management practices and therefore what actual stream concentrations to and from the farm boundary would be.

Table 1 (Farm A) and Table 2 (Farm B) shows the differences in stream loads at entry and exit points which overlap between outputs from the farm in isolation versus the farm in the context of its contributing upstream/uphill areas. Loads and




concentrations are generally lower when just the buffer around the study areas are used compared to when the upstream contributing areas are included. By modelling only the study area in isolation with a 2-pixel buffer around it, the modelled loads and concentrations may not be truly representative of the loads and concentrations occurring at the exit points of the farm, most crucially the main outlet. The impact of the farm on water quality is thus underestimated and measurements of

concentration at the outlet may be higher due to the contribution of uphill areas. When the "net impact" of the farm is better understood, this information affects land management decisions around where to put interventions on the farm, or if interventions are needed further upstream and possibly on the property of other landowners or landscape management authorities.

*Table 1. Nitrogen loads (kg), concentrations (mg L⁻¹) and streamflow (L sec⁻¹) at entry and exit points along the boundary of Farm A using LUCI v0.9 (Figure 11). Only those entry and exit points which coincide across both the study area \ and the upstream contributing area are shown.*

| Study area only | | | | | Upstream contributing area | | | | |
|---|---|---|---|---|---|---|---|---|---|
| Point # | Type | N load (kg) | N conc (mg L⁻¹) | Flow (L sec⁻¹) | Point # | Type | N load (kg) | N conc (mg L⁻¹) | Flow (L sec⁻¹) |
| 5 | Exit | 31.34 | 0.72 | 1.38 | 5 | Exit | 48.91 | 1.14 | 1.36 |
| 1 | Exit | 73.66 | 1.10 | 2.13 | 1 | Exit | 75.55 | 1.12 | 2.13 |
| 2 | Main exit | 138.30 | 0.26 | 16.71 | 2 | Main exit | 1131.64 | 0.53 | 67.57 |

*Table 2. Nitrogen loads (kg), concentrations (mg L⁻¹) and streamflow (L sec⁻¹) at entry and exit points along the boundary of Farm B using*
*LUCI v0.9 (Figure 13). Entry and exit points which coincide across both the study area only output and the upstream contributing area output are shown, but the main exit when using input from the contributing area is also included.*

| Study area only | | | | | Contributing area | | | | |
|---|---|---|---|---|---|---|---|---|---|
| Point # | Type | N load (kg) | N conc (mg L⁻¹) | Flow (L sec⁻¹) | Point # | Type | N load (kg) | N conc (mg L⁻¹) | Flow (L sec⁻¹) |
| 1 | Exit | 47.16 | 0.68 | 2.21 | 1 | Exit | 267.09 | 0.81 | 10.43 |
| 2 | Exit | 103.28 | 1.70 | 1.93 | 2 | Exit | 103.29 | 1.70 | 1.93 |
| 3 | Exit | 152.03 | 1.96 | 2.46 | 3 | Exit | 171.40 | 1.61 | 3.37 |
| 4 | Main Exit | 775.60 | 1.61 | 15.22 | 4 | Exit | 805.26 | 1.67 | 15.32 |
| | | | | | 7 | Main exit | 771.05 | 0.81 | 30.00 |

At the farm scale, the interconnectedness of the farm to the surrounding areas and the possible impacts on water quality show the importance of considering the entry/exit points going into and out of the farm. Understanding how streams move, exit, and
enter a catchment or subcatchment is also important for management activities. The following case studies show how *LUCI-EntEx v1.0* can be used at the catchment scale in order to identify the main outlet of the catchment, entry/exit points, and show possible inconsistencies between the input DEM and stream network.





At the catchment scale, the algorithm was able to identify the main outlet in the correct location for the Mangatarere using both the 5m and 15m DEMs (Figure 15). If the study area is a self-contained catchment, then no entry points are expected to be found. If any entry points are identified, the algorithm will warn the user that entry points have been found even though the study area is a self-contained catchment. The identification of entry points may occur due to inconsistencies between the DEM

5 and the stream network The *Preprocess DEM* tool addresses these inconsistencies using stream burning and other hydrological GIS tools. However, there could also be inconsistencies between the study area mask and the input DEM; for example a political/legal "catchment boundary" recognised by a regulatory authority may have been derived using different elevation and processing than the DEM. Therefore, depending on the input data and study area characteristics, not all inconsistencies may be reconciled.

As with delineating streams from a DEM, catchment boundaries may be delineated differently depending on the input data and choices around the parameters input to the hydrological tools. The "legal" administrative boundaries of catchments may also differ from the hydrological boundaries created from catchment delineation. The identification of exit points from the catchment can help with land management decisions since activities within the catchment would affect the areas downstream

15 from these exit points even if they are beyond the catchment boundary.

Figure 15 shows the stream network and entry/exits for the Mangatarere catchment identified by *LUCI v0.9*. Although not shown in this paper, the stream network identified by *LUCI-EntEx v1.0* is broadly the same and the outlet points are in the same place. In terms of the differences in results produced by *LUCI-EntEx v1.0*, the run using the 15m DEM does not have

20 any entry/exit points other than the main outlet, and the run using the 5m DEM has exit points in the same places but with one less exit point compared to Figure 15b. For the entry point found using the 15m DEM, Figure 16 shows that a stream has initiated at the boundary of the study area mask, travels along the boundary, and re-enters the study area. The distance between this pair of entry/exit points is above the distance threshold, hence they were not marked as spurious. Therefore, there is still room for improvement in the code to account for edge effects like this in large watersheds.





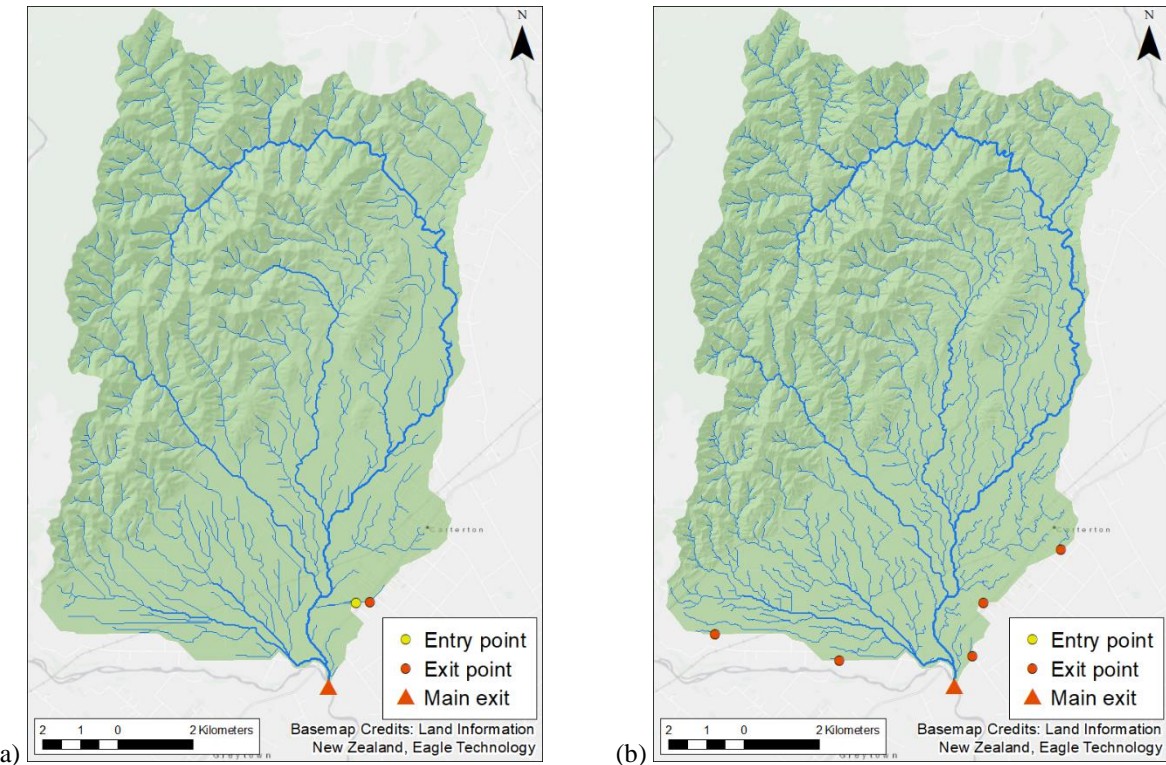

*Figure 15. Stream entry/exit points for the Mangatarere catchment using the (a) 15m DEM and (b) the 5m DEM generated using the LUCI v0.9 code.*



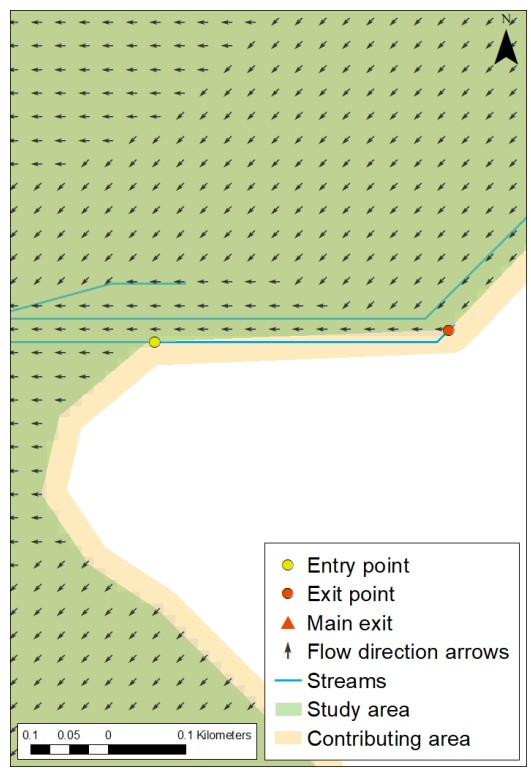

*Figure 16. Close up view of stream entry/exit points occurring on the boundary of the Mangatarere catchment when using the 15m DEM and LUCI v0.9.*

These edge effects, the effect of the inconsistencies between DEMs and stream networks, and the issues with determining flow direction in flat areas are even more pronounced in the application on the Piako catchment (Figure 17). More entry points were identified, mainly due to streams initiating near the boundary of the study area (Figure 18a) or streams crossing along the boundary where the distance between the entry/exit points are higher than the distance threshold (Figure 18b). This may be due to the Piako having very flat areas where the D8 algorithm may not be enough to accurately capture the flow of water

(Zhang et al., 2017). Further improvements to the code may include the implementation of more complex flow direction procedures such as the D-infinity or Multiple Flow Direction (MFD) algorithms (Quinn et al., 1991; Tarboton, 1997). The flow networks on agricultural areas have also been modified due to the presence of canals and drains. These modifications may be reflected in the DEM or the stream network, adding another complication to the process of reconciling these inputs. When the *LUCI-EntEx v1.0* code was used, all of the entry/exit points and outlet locations were the same as when the *LUCI*

*v0.9* code was used.





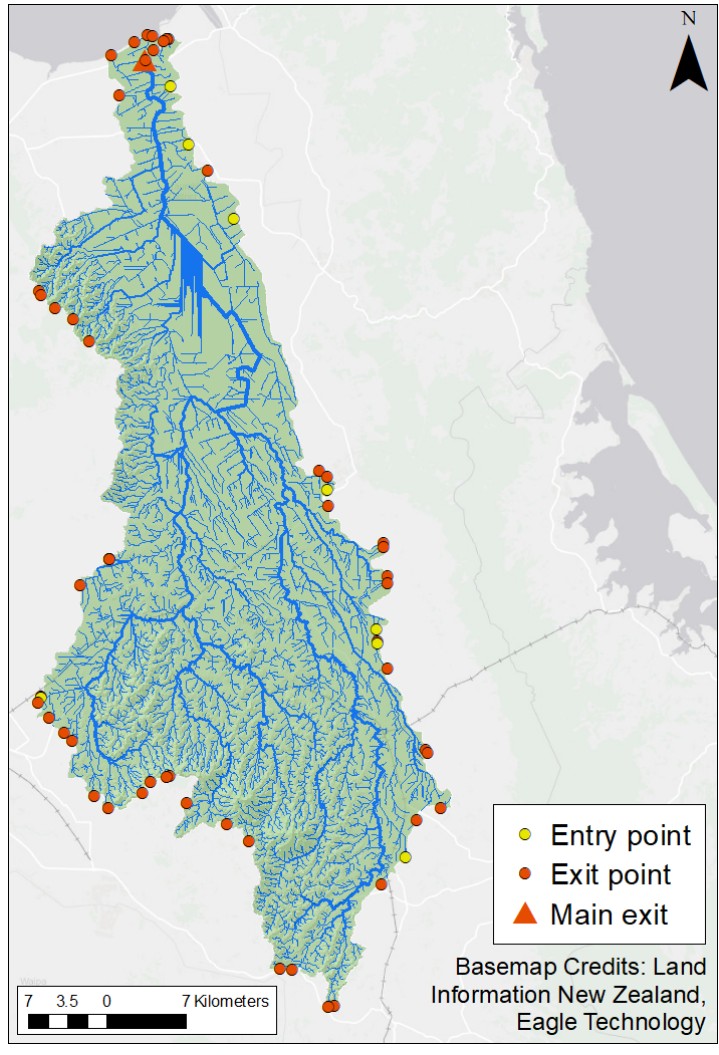

*Figure 17. Stream entry/exit points for the Piako catchment generated using LUCI v0.9.*



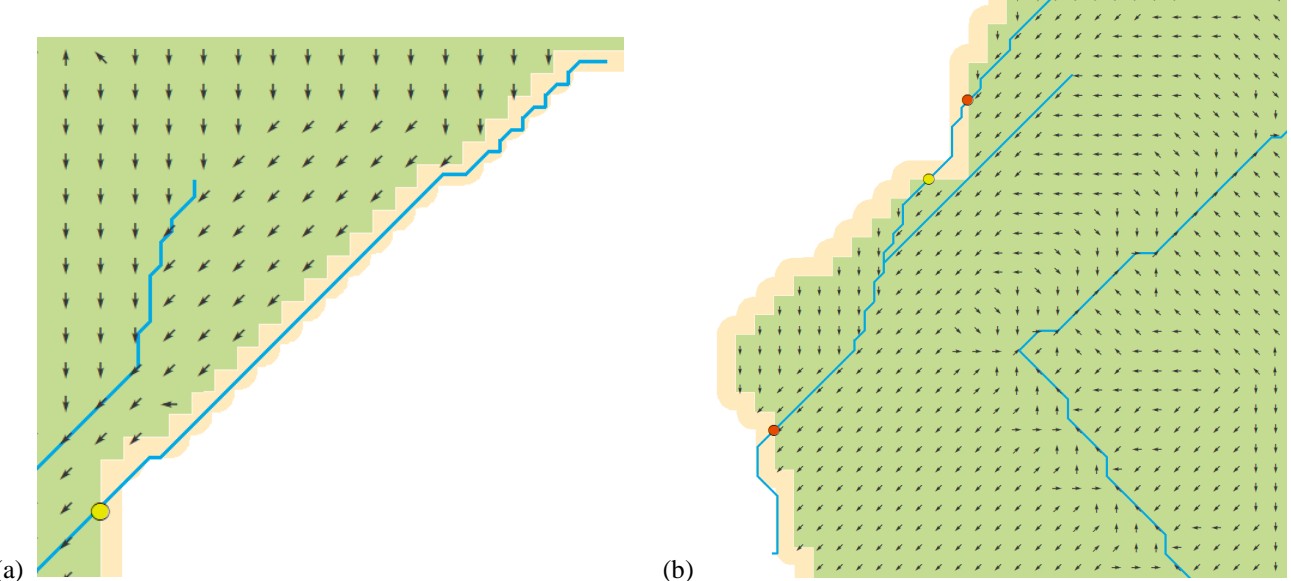

*Figure 18. Close up view of stream entry/exit points occurring on the boundary of the Piako catchment generated using LUCI v0.9.*

**Terrestrial entry/exit procedure**

A further complication at the farm scale is the terrestrial part of the landscape where nutrients, sediment or other mass may enter the study area from neighbouring properties through terrestrial pathways. Many farm boundaries do not coincide with
actual catchment boundaries and therefore nutrients or other mass may cross into the study area and contribute to in-stream loads and concentrations. Conversely, loads generated within the farm may exit the farm via terrestrial nutrient pathways that are directed into neighbouring properties.

By understanding the terrestrial pathways of nutrients and sediment in relation to the farm's location and neighbouring
properties, landowners may be able to get a clearer picture of how onsite practices influence the water quality of nearby properties, and how practices on nearby properties may affect their onsite water quality. Figure 19 and Figure 20 show the terrestrial entry/exit procedure when applied to Farm B. In the northern borders of the farm, most of the terrestrial flow is going into the study area or along a ridgeline (Point A, Figure 19), while in the southern border of the same area (Point B, Figure 19) flow is going out, and terrestrial flow is predicted to be going into the farm on the eastern border of the western
side of the farm (Point C, Figure 19). When checking these results with those with local knowledge of the farm, the overall patterns of flows entering and exiting are agreed to be broadly correct. However, some flow has been predicted to occur on boundaries which are largely on a ridge line, including a segment which is a road boundary along a ridgeline. This speaks to the issue of artefacts in the digital elevation data and/or inconsistencies in legal/political versus "real" boundaries compromising the integrity of results. It shows the importance of ground-truthing results with local stakeholders where



possible, and perhaps including consideration of magnitude of flow and uncertainty in identified flow directions, the former decreasing and the latter increasing as slopes approach zero.

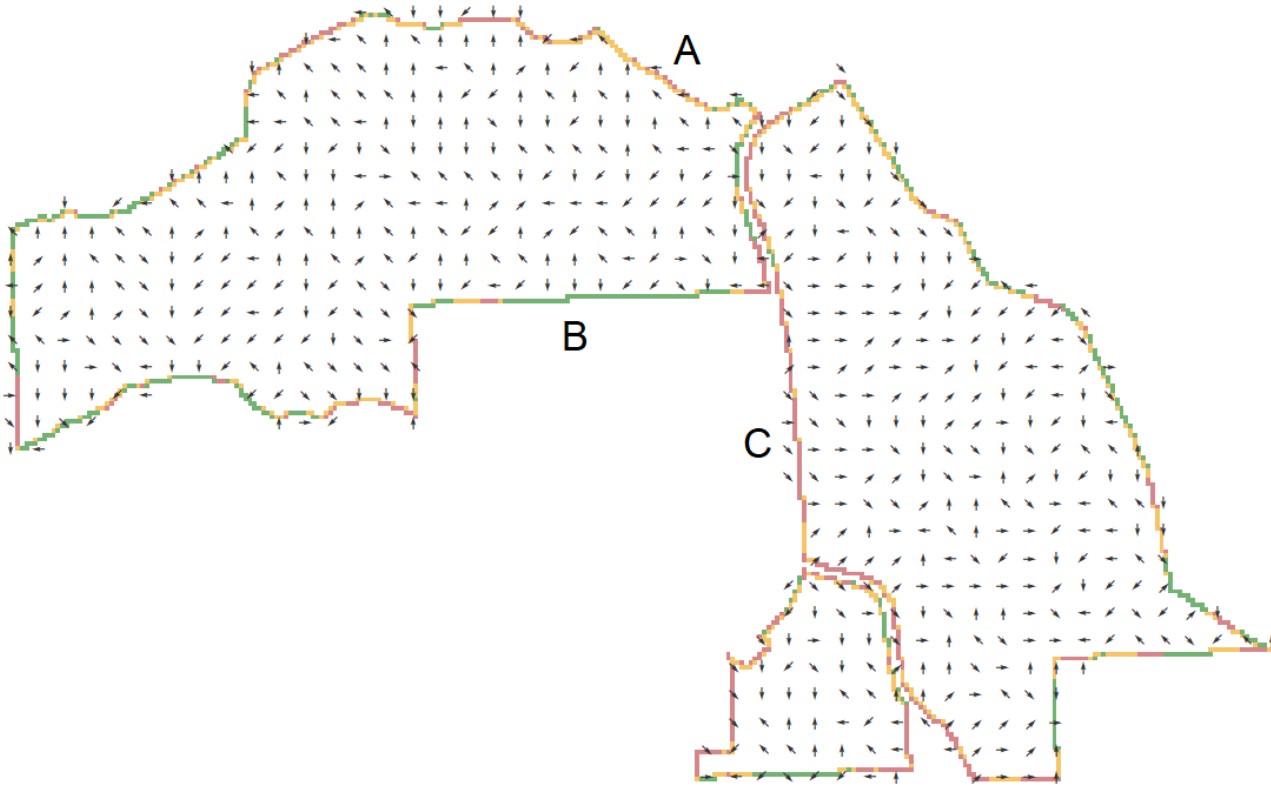

*Figure 19. Outermost cells of the study area mask with the direction of flow (black arrows) displayed. Green cells indicate where flow is leaving the study area. Red cells indicate where flow direction is toward the study area mask and orange indicates the flow going along ridges. The flow direction arrows outside the study area mask are due to the 2-pixel buffer applied in the DEM preprocessing stage.*

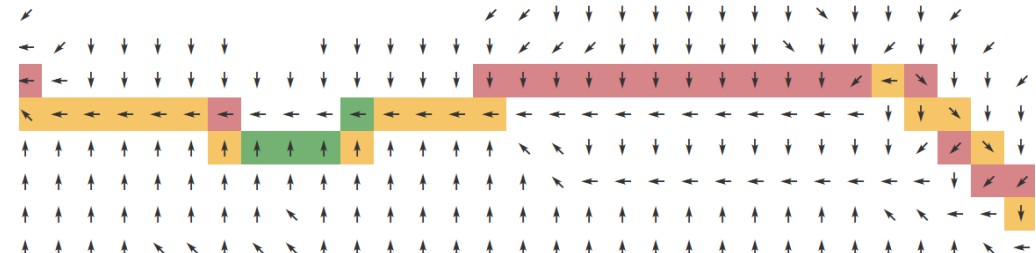

*Figure 20. Close up view of the outer most cells of the study are mask. Green cells indicate where flow is leaving the study area. Red cells indicate where flow direction is toward the study area mask and orange indicates flow going along ridges.*





## 5 Conclusion

The algorithm described in this paper allows for the automatic identification of stream entry and exit points at intersection locations along the study area boundary and identifies areas where terrestrial flow enters or leaves the study area. Although such identification is generally possible by eye, the automation of this procedure is complex as demonstrated by the number

of GIS calculations required to cover the range of complexities encountered when processing discrete gridded topographical data (a DEM) and accounting for boundary/edge effects etc. This algorithm is expected to be of value for a range of models and policy decisions surrounding water, nutrients, sediments, etc. At the farm scale, it is important to understand not only where flow exits the farm, but the areas where inputs from uphill/upstream sources can affect onsite and exiting water and sediment fluxes, and water quality. By calculating mass and concentrations of substances of interest at the entry and exit points

of any given area, it becomes possible to quantify impacts of management attributable to that area only, and distinguish local management effects from effects of upstream and uphill management. At the catchment scale, the exit points identify outlets of the "catchment", which can include streams directly entering lakes and oceans as well as the main outlet. The algorithm is also valuable in elucidating any inconsistencies between the DEM, stream network, and study area mask that must be addressed. This algorithm is supported both as stand alone, freely available and open source code (*LUCI-EntEx v1.0*), and it

is also embedded within the larger LUCI framework (*LUCI v0.9*) for use in understanding water quality at entry and exit points from land parcels of any relevant size under different scenarios. Both this standalone tool and the larger LUCI framework are undergoing further development to improve its accuracy, applicability, and address limitations identified in this application and other applications. In the farm-scale case studies, the code can remove spurious entry/exit points that occur along the boundary due to a stream crossing in and out of it by assessing the distance threshold between points. For smaller study areas

(i.e. farms), a distance threshold of 100 spatial reference units may be appropriate, but at larger scales such as the catchment scale, this distance threshold may be modified to reflect study area characteristics. Further work intends to include more sophisticated ways of removing spurious entry/exit points, more appropriate ways of determining the flow direction in topographically complex and/or flatter study areas, and refining the algorithm to be of further value to coastal study areas.

*Code and data availability.* The *LUCI-EntEx v1.0* code is available on GitHub (https://github.com/lucitools/LUCI_EE). A tutorial with sample data (.zip format) is available through the LUCI website (https://lucitools.org/luci-tutorials/running-lucientex-v1-0). The digital elevation model (DEM) raster files and stream network vector files provided as example data are clipped from freely available regional and national New Zealand datasets licenced under Creative Commons 3.0. The original

DEM raster was produced by Columbus et al. (2011) and the stream network vectors by Snelder et al. (2010). Installation and user instructions are available on the readme.md file included in the GitHub repository. If this paper is accepted by GMD, we will follow journal best practice guidelines and archive the version and example data as published in one of the recommended open source repositories. The full *LUCI v0.9* code and data used to demonstrate some of the utility of this algorithm cannot at



this point be made available for open redistribution due to third-party data and licensing restrictions; please visit www.lucitools.org for further and up to date information on its availability.

*Author contribution.* All authors have significantly contributed to the work described in this manuscript. The original concept
and high level algorithm for the stream entry/exit points was formulated by BJ, BJ and KM formulated the initial implementable algorithm, and KM implemented the original code. The algorithm and code were further amended and updated by RB with input from BJ. DM collated the first full draft of this manuscript, and BJ and RB added additional text and other material as the algorithm and code evolved.

*Competing interests.* The authors are not aware of any competing interests.

*Acknowledgements.*

We are grateful to Waikato Regional Council, Greater Wellington Regional Council, NZ Ministry for the Environment, Manaaki Whenua/ Landcare Research and Ravensdown for data provision along with broader support and encouragement.
Ravensdown staff provided valuable input into why an algorithm of this type is necessary to aid local management, and Alister Metherell from Ravensdown also provided constructive comments that improved this manuscript. We are also very grateful to the farm owners and employees from our study site farms. They not only shared their data and insights, but were generous in showing us around their farms to help us better understand some of the differences between the digital /modelled and "actual" landscape realities.

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
