# Peer review of "LUCI-EntEx v1.0: A GIS-based algorithm to determine stream entry and exit points at boundaries of any given shape."

_Geoscientific Model Development, 2020_

## Short Comment (SC1) · 21 Dec 2020

Dear authors,

as executive Editor of GMD, I am writing this comment to make you aware of a misinterpretation of our "best practive" rules as you name it. In your code availability section you write "If this paper is accepted by GMD, we will follow journal best practice guidelines and archive the version and example data as published in one of the recommended open source repositories". This is not exactly to the point, as we require the authors to archive the data **before** the discussion phase. Therefore please write as soon as possible a comment stating the archive place of your code.

[Figure]

Best regards,

Astrid Kerkweg
* * *

---

## Referee Comment (RC1) · Anonymous Referee #1 · 11 Jan 2021

General comments

This paper described an algorithm to identify the entry/exit points at boundaries with any given shape. The LUCI-EntEx v1.0 and LUCI v0.9 were developed to automate the process for preprocessing the DEM, identify the stream entry/exit points, and determine the terrestrial entry/exit procedure at boundaries. The use cases for both farm and catchment scales were implemented to validate the GIS tools.

The major contribution of this work is the proposed algorithm to identify the stream entry/exit points at boundaries and the code implementation, which is a novel idea and not available in many of the existing GIS tools. The automation of the DEM preprocess-

ing and identification of the terrestrial entry/exit procedure with existing tools/algorithm also contribute a bit to simplify the work for data preprocessing and analysis. This paper presented the workflows and algorithm in a logical and clear way. The findings that the different contributing areas (buffer or upstream contributing area) in a farm scale will lead to different results are also informative. Detailed comments for this paper are listed below.

Detailed comments

1. pg 2 line 15: The paper only cited Arc Hydro (Maidment 2002) as GIS tools for catchment delineation. It would be better to provide a bit more description as a brief literature review of various GIS tools and algorithms for catchment delineation. This will help people especially for those who are not familiar with the GIS tools for catchment delineation to get a better sense how your work fit into a bigger context and the contribution of your work.

2. pg 5 fig 1: For the step "Burned DEM" -> "Calculate flow direction", this should be "hydDEM" ->"Calculate flow direction". The flow direction is calculated using the DEM of which the sinks are filled.

3. pg 7 line 2-3: Please provide a bit more information to explain how the "not determined points" are handled. Are they discarded directly?

4. pg 7 line 11: "is the located" should be "is located".

5. pg 9 fig 4: At the bottom of the figure, the flowchart has a bit overlap for the part "Is starting point inside study area mask" (the left one) and its option "N".

6. pg 15 line 24: What is NZSoS? Please spell its full name and put the abbreviation in parentheses.

7. pg 16 fig 10: Please add a scale bar and a legend in the map. It is also suggested to change the way to show the locations of the 4 study areas. In this map, the farm areas are big green points (hexagon) and the catchments are showing the actual shapes. It
is a bit misleading for readers to think the big green points for the farms are the actual shapes at the first glance of the map.

8. pg 18 fig 12 (a): I suppose both LUCI-EntEx v1.0 and LUCI v0.9 are using the same processed DEM data with the same small buffer area. In fig 11 (a), the caption is "buffer(beige)", will this be the same for the fig 12 (a) caption? Instead of "buffer (not shown)", it may be buffer (beige).

9. pg 19 line 13: In fig 14(b), it is number 15 not 14.

10. pg 20 fig 14 (a): This is the similar issue as fig 12 (a). The caption "buffer (not shown)" may need correction.

11. pg 21 table 1: Remove "\" in the caption for " study area \ and the upstream. . .".

12. pg 22 line 5: Missing a period "." before the sentence " The Preprocess DEM tool. . .".

13. pg 26 fig 18: Please add caption info for plot (a) and (b) separately after the general description of the figure.

---

## Author Comment (AC1) · 10 Mar 2021

Thank you very much for the constructive comments. We were intending to address them in a combined response to reviewers, but given understandable delays in the second review coming in given global circumstances, have decided to give an interim response now.

Regarding your comments on the contribution, we completely agree with those and that the first and main contribution you note is is novel and valuable, but also that the further automation and error/issue checking in standard hydrological GIS pre-processing we have implemented may be useful to other researchers although we certainly do not

claim that aspect to be a scientific advance.

We will incorporate most of your suggestions into our revised manuscript once the second review is secured. Although we agree your point 7) would be ideal, and we will add the scale bar and legend, there are some slight privacy issues around the farm locations so we can't fully address the second part of your suggestion.

To address the most substantive of your detailed points, here is the revised, expanded discussion we intend to insert in the revised manuscript detailing the background and start of the art in catchment delineation tools:

Revised paragraph(s) re: catchment delineation tools

Creating a hydrologically and topographically consistent digital elevation model (DEM) with an appropriate stream network is an important part of modelling landscapes in many hydrological applications. Reasonably accurate identification of the stream network is particularly important in understanding transport of water, sediment, nutrients and biomass through a landscape. Terrain analysis of topography data is used to produce catchment boundaries and surface water features through removing pits, calculating drainage directions and flow accumulations, and defining stream channels where pixels exceed an accumulation threshold (Tarboton et al., 1991). The manual process is tedious and has given rise to GIS toolboxes to conduct terrain analysis with some automation to process inputs to spatially model the movement of water and other mass through a landscape (Maidment, 2002).

Popular algorithms for terrain analysis are now commonly embedded in common GIS platforms such as ArcGIS/ArcPro, QGIS, GRASS GIS, SAGA GIS etc. Some stand-alone tools also exist such as DelineateIT (Sharps et al., 2020), and TauDEM (Tarboton et al., 2009; Tesfa et al., 2011). These tools all are capable of running established hydrological geoprocessing steps; preprocessing a DEM through filling sinks and removing pits, calculating estimates of flow direction and accumulation, extracting stream networks, and producing catchment boundaries. Standalone tools such as DelineateIT and TauDEM include some more complex tools to enable more sophisticated processing. Innovations in these and other research focussed on improving catchment delineation and flow processing have led to the availability of more complex algorithms able to better partition flow, often utilising parallel processing to increase computational efficiency (Tesfa et al., 2011; Haag et al., 2018; Sit et al., 2019). Other research has also utilised complementary satellite data to directly extract surface water features and catchment boundaries (Li et al., 2019) (noting that many DEMs themselves are partly constructed from satellite information among other data sources).

However, these tools have been generally designed and used to understand flow pathways within an isolated catchment or subcatchment, where there is no water transfer from outside the catchment boundary and there is often only one significant outlet to consider. Complexities arise when the area to be modelled covers only part of a catchment or encompasses several subcatchments with multiple entry and exit points along its boundary. This is the case with many farms, forestry units, and other land management units that have been defined according to administrative boundaries rather than natural catchment and subcatchment boundaries.

References Haag, S., Shakibajahromi, B., & Shokoufandeh, A. (2018). A new rapid watershed delineation algorithm for 2D flow direction grids. Environmental Modelling and Software, 109(August), 420–428. https://doi.org/10.1016/j.envsoft.2018.08.017. Li, L., Yang, J., & Wu, J. (2019). A method of watershed delineation for flat terrain using Sentinel-2A imagery and DEM: A case study of the Taihu basin. ISPRS International Journal of Geo-Information, 8(12). https://doi.org/10.3390/ijgi8120528. Sharp, R., Douglass, J., Wolny, S., Arkema, K., Bernhardt, J., Bierbower, W., Chaumont, N., Denu, D., Fisher, D., Glowinski, K., Griffin, R., Guannel, G., Guerry, A., Johnson, J., Hamel, P., Kennedy, C., Kim, C.K., Lacayo, M., Lonsdorf, E., Mandle, L., Rogers, L., Silver, J., Toft, J., Verutes, G., Vogl, A. L., Wood, S., & Wyatt, K. (2020). InVEST 3.9.0.post25+ug.g6361440 User's Guide. The Natural Capital Project, Stanford University, University of Minnesota,

The Nature Conservancy, and World Wildlife Fund. Retrieved February 2nd, 2020 from https://storage.googleapis.com/releases.naturalcapitalproject.org/invest-userguide/latest/index.html. Sit, M., Sermet, Y., & Demir, I. (2019). Optimized watershed delineation library for server-side and client-side web applications. Open Geospatial Data, Software and Standards, 4(1). https://doi.org/10.1186/s40965-019-0068-9. Tarboton, D. G., Bras, R. L., & Rodriguez‐Iturbe, I. (1991). On the extraction of channel networks from digital elevation data. Hydrological Processes, 5(1), 81–100. https://doi.org/10.1002/hyp.3360050107. Tarboton, D., Schreuders, K., Watson, D., & Baker, M. (2009). Generalized terrain-based flow analysis of digital elevation models, 18th World IMACS Congress and MODSIM09 International Congress on Modelling and Simulation, ed. R. S. Anderssen, R. D. Braddock and L. T. H. Newham, Modelling and Simulation Society of Australia and New Zealand and International Association for Mathematics and Computers in Simulation, July 2009, p.2000–2006. Retrieved February 2nd, 2020 from http://www.mssanz.org.au/modsim09/F4/tarboton_F4.pdf. Tesfa, T. K., Tarboton, D. G., Watson, D. W., Schreuders, K. A. T., Baker, M. E., & Wallace, R. M. (2011). Extraction of hydrological proximity measures from DEMs using parallel processing. Environmental Modelling and Software, 26(12), 1696–1709. https://doi.org/10.1016/j.envsoft.2011.07.018.

Best wishes, the authors.

---

## Referee Comment (RC2) · Anonymous Referee #2 · 5 May 2021

The paper describes the LUCI-EntEx v1.0 algorithm for the preprocessing of DEMs, automatic identification of intersections between the river network and the boundary of the study area and classification into entry and exit points. The algorithm also allows to determine if the overland flow on the boundary cells enters or leaves the catchment. The algorithm is in the form of a toolbox for ArcGIS and has been developed both as a standalone tool as well as embedded into LUCI v0.9. The paper presents the application of the toolbox to two farms and two catchments in New Zealand, and compares the standalone version vs the version embedded in LUCI v0.9.

The tool for identification of entry/exit points is novel and can be useful to simplify

geospatial analyses. The paper is overall well written, describes the algorithm in detail and presents the applications in a logical order. However, I think that the structure of the paper should be improved and that several aspects should be clarified, to clearly state which parts of the code are new and which have been taken from previous models. Also, some analyses should be improved, especially that of the net impact of the farm on the water quality, and the scientific context of the proposed applications clarified. It is not stated which research questions the authors want to address with the comparison between LUCI v0.9 and LUCI-EntEx v1.0, and between the 15m and 5m DEM application on the Mangatarere catchment, and which conclusions are drawn from them. I think that the paper would also strongly benefit from a discussion section separated from the results, where these scientific questions are addressed.

**Main comments:**

- It is not clearly stated if the "pre-processing DEM" algorithm (Fig. 1) has been developed in the context of this work, or has been taken from the LUCI v0.9 model.

- LUCI EntEx v1.0 is freely available and open source, but it requires ArcGIS which is not free. I think this diminishes the meaning of LUCI EntEx v1.0 being open source. What is the reason for developing the LUCI-EntEx v1.0 for ArcGIS and not QGis, for example? To which extent can the algorithm be adapted to QGis?

- Fig. 11 to 14 show that the upstream contributing areas need to be modelled in order to have accurate results from the algorithm, i.e. the algorithm is sensitive to the dimension of the buffer around the study area. Isn't this result expected? What is the scientific question you want to address by comparing the application including the upstream area (panels b) vs the 2-pixel buffer (panels a)? I already expect that omitting the upstream area will provide a much less precise solution. Is there any case in which omitting the upstream area could still be a meaningful choice?

- For the farm applications (Fig. 11 to 14) could you provide a map of the entry/exit

points manually identified from the on-ground stream network, to validate the algorithm and the thesis that the panels (b) of the figures are more correct? It is easy to expect that the algorithm is helpful in identifying entry/exit points from large catchments, but does it really provide an advantage to manual mapping in the case of farms? Especially given the sensitivity to the modelled upstream area, and likely other parameters too.

- Table 1 and 2: in lines 13-16 p. 20 the authors say that they want to estimate the net impact of the farms on the nitrogen loads, but in Table 1 and 2 only the loads at the exit points are shown, which is not sufficient to quantify such impact. To quantify the impact, the loads at the entry points should also be reported, and I also suggest to add a line at the end of the table to show the net impact of the farm (difference between total exit loads and total entry loads). Only with these data the underestimation of the net impact when not considering the upstream contributing areas can be quantified.

- It is not explained what the authors want to show with the comparison of the model application with the 15m and 5m DEM for the Mangatarere catchment in Section 3 (Fig. 15), as well as what are the conclusions drawn from it.

- Theoretically speaking one should not expect to find entry/exit points along the boundary in the case of self-contained catchments, however there can be mismatch between catchment masks and physical boundary, as the authors mention at page 22. Could you provide some examples of concrete situations/conditions in which such points should be expected? And therefore, when the application of LUCI-EntEx v1.0 at the catchment scale is recommended.

- I think that some context about the LUCI framework and its relationship with LUCI-EntEx v1.0 is missing. Currently, it is partially explained at page 14, but still several aspects are missing: what are specifically the differences between LUCI v0.9 and LUCI-EntEx v1.0 in the derivation of the stream network? What is the reasons for such different choices in LUCI-EntEx v1.0? At the moment, the motivation for the comparison between the two models is not explained. Could you discuss how these affect

the stream networks shown in Fig. 11 to 14? What should the user keep in mind when choosing to use LUCI v0.9 or LUCI-EntEx v1.0?

- What is the advantage of having LUCI-EntEx v1.0 as a standalone tool compared to the same algorithm embedded in LUCI v0.9?

**Specific comments**

**Introduction:**

- I think the clarity of the paper would benefit from a more structured anticipation of the content of the sections at the end of the introduction: how is the new algorithm used to estimate the net impact of farms, mention that you will look at nitrogen concentration with LUCI and explain why you chose this pollutant, introduce that you will compare the application including the upstream area VS small buffer, and the 15m and 5m DEM, and why such applications are of interest.

- Also introduce briefly what LUCI is and explain that in this paper a part of that code will be used and further developed to create a standalone tool, and the new component has also been embedded in LUCI. And that you will provide a comparison between the two

- Add a literature review to support your thesis that this algorithm complements the available GIS tools, if and which other similar tools are present, and what is the advantage provided by this new algorithm.

**Section 2:**

- Line 15 page 3: here it should be specified that the input stream network is an independent ground-measured one. I did not understand the meaning of having it as an additional input file, given that it can be derived from the DEM, until page 4.

- Fig 1: in the caption it is said that the procedure has been taken from Maidment

(2002). Please cite this source also in the description of the preprocessing of the DEM in the main text. Now it is present in line 17 p. 4, but not as the source of the preprocessing procedure.

- Figure 2: why is there (a) in front of "Burn streams into the DEM"? If it is to say that this figure is a sub-process of Figure 1, I would add the same (a) after "Burn streams into the DEM" in the yellow box in Figure 1. If this is done, please also do the same for Fig. 3 and 4 for the "Find if intersection points are entry or exits points"

- All flow charts: adapt the dimension of the rectangles/diamonds to the length of the sentence to improve readability and to allow writing more complete sentences. Also the figure resolution should be increased.

- Fig 4: right arrow starting from diamond "Is starting point inside study area mask" is misaligned

- Fig 8: write FAC as fac (italics) coherent with the rest of the text and use the caption to provide a summary of the described process

- Fig. 9: use the caption to provide a summary of the described process

**Section 3:**

- Lines 24-28 p. 14: this summary should be come earlier in the text (at the end of introduction or the beginning of the methods)

- Line 27 p. 14: Nitrogen is mentioned here for the first time. It would be better to introduce it earlier and motivate why it is chosen for the modelling

- Lines 29, 30 p. 15: is that the procedure described in section 2 "preprocessing of the dem"? If so, refer to the section for clarity

- Line 30-31 p. 15: I do not understand what you did here. Is that a comparison of LUCI-EntEx v1.0 with an established ArcGIS tool? If so, how does the ArcGIS tool compare with LUCI?

**Section 4:**

- Line 10 p. 16: "Applying the methodology above", refer to the section where the methodology is described

- Line 5 p. 17: "reflects the fact that the accumulation ...", add "that"

- Fig 11, 12: remove the black lines around the figures and overlap the maps of the farm to the hillshade of the DEM as in Fig. 13 and 14

- Line 1 p. 18: "there can be instances where another tributary enters converges with the river just outside the farm boundary". Did you mean "exits, "?

- Fig. 13: can you repeat in the caption the description of point 1 and point 10 in panel (b), and specify that point 5 in panel (a) is located after the convergence of the streams? Alternatively try to enlarge the figures to show it, for example by saving space removing one of the legends.

- Fig. 14 has the same legend as Fig. 13. You could instead use it to highlight some differences between Fig. 14 and Fig. 13.

- Line 2 p. 21: it is the first time that it is mentioned that the buffer is 2-pixels. Please specify it also in Section 3 in the description of the numerical applications

**Code and data availability:**

- Line 27. The link to the tutorial doesn't work